# Organo-Montmorillonite Modified by Gemini Quaternary Ammonium Surfactants with Different Counterions for Adsorption toward Phenol

**DOI:** 10.3390/molecules28052021

**Published:** 2023-02-21

**Authors:** Ran Wei, Yuanhua Mo, Duojiao Fu, Hongqin Liu, Baocai Xu

**Affiliations:** School of Light Industry, Beijing Technology and Business University, No. 11 Fucheng Road, Haidian District, Beijing 100048, China

**Keywords:** modified montmorillonite, Gemini surfactant, counterions, adsorption, phenol

## Abstract

The discharge of industrial phenol pollutants causes great harm to the natural environment and human health. In this study, phenol removal from water was studied via the adsorption of Na–montmorillonite (Na–Mt) modified by a series of Gemini quaternary ammonium surfactants with different counterions [(C_11_H_23_CONH(CH_2_)_2_N^+^ (CH_3_)_2_(CH_2_)_2_ N^+^(CH_3_)_2_ (CH_2_)_2_NHCOC_11_H_23_·2Y^−^, Y = CH_3_CO_3_^−^, C_6_H_5_COO^−^ and Br^−^, 12–2–12·2Y^−^]. The results of the phenol adsorption indicated that MMt–12–2–12·2Br^−^, MMt–12–2–12·2CH_3_CO_3_^−^ and MMt–12–2–12·2C_6_H_5_COO^−^ reached the optimum adsorption capacity, which was 115.110 mg/g, 100.834 mg/g and 99.985 mg/g, respectively, under the conditions of the saturated intercalation concentration at 2.0 times that of the cation exchange capacity (CEC) of the original Na–Mt, 0.04 g of adsorbent and a pH = 10. The adsorption kinetics of all adsorption processes were in good agreement with the pseudo-second-order kinetics model, and the adsorption isotherm was better modeled by Freundlich isotherm. Thermodynamic parameters revealed that the adsorption of phenol was a physical, spontaneous and exothermic process. The results also showed that the counterions of the surfactant had a certain influence on the adsorption performance of MMt for phenol, especially the rigid structure, hydrophobicity, and hydration of the counterions.

## 1. Introduction

Phenol and its derivatives frequently appear in industrial wastewater as harmful pollutants with a strong corrosive effect [1,2]. Therefore, the discharge of industrial phenol pollutants causes great harm to the natural environment and can significantly pollute the water and atmosphere [3]. In addition, phenol also has adverse effects on the human body [4]. It has a strong corrosive effect on human skin and mucous membranes, and may inhibit the central nervous system, leading to liver and kidney damage. At present, various technologies have been proposed to remove phenols from industrial wastewater, such as distillation, extraction, adsorption, chemical oxidation, electrochemical oxidation, enzymatic treatment, membrane technology, biodegradation, photocatalytic degradation, etc. [5,6,7,8]. Among them, distillation and electrochemical oxidation require high energy consumption, chemical oxidation has the disadvantage of being cost-ineffective due to a continuous supply of chemical and energy, enzymatic treatment is highly dependent on reaction conditions, fouling is the main parameter that limits the applications of membrane technology, biodegradation is suitable for the treatment of low-concentration phenol, and in photocatalytic degradation, the extra step in recovering the catalyst could increase the operating cost [9,10]. Therefore, adsorption is widely used due to its low cost, high efficacy, and ecofriendly and easy operation [8,9,10,11]. In recent years, with environmental and economic motivations, the research and development of environmentally safe, highly effective, and low-cost adsorbents to eliminate various pollutants from water and wastewater has attracted extensive attention [12,13,14,15,16,17].

Natural Mt is a kind of hydrous aluminosilicate mineral, which is one of the most popular clay precursors. Natural Mt has been widely adopted because of its high surface area, certain adsorption capacity, and low cost [18]. However, due to the low organic content and poor hydrophilicity of natural Mt, its application in wastewater treatment is limited. However, it is worth noting that natural Mt has the ability of isomorphic substitution; that is, the compensating cations between clay mineral layers can be replaced by organic cations to form organic intercalation [19]. Therefore, cationic surfactants can be used to modify natural Mt through ion exchange or intermolecular interaction to obtain modified organic Mt, which can greatly increase the interlayer spacing and hydrophilicity of Mt, thereby improving the adsorption effect of organic pollutants [20].

At present, quaternary ammonium cationic surfactants are the most widely used organic modifiers [5]. Comparing with monomeric surfactants, Gemini quaternary ammonium surfactants have exceptional properties, such as low critical micelle concentration (CMC), excellent wetting ability, superior aggregation behaviors, dispersion stabilization, etc. [21,22]. In addition, several studies proved that Mt modified by Gemini quaternary ammonium surfactants exhibited better efficacy than that modified by monomeric surfactant in removing organic pollutants from wastewater [5,23,24]. The adsorption performances of modified Mt, such as adsorption capacity, equilibrium time, mechanism, and regeneration, are closely related to specific differences among Gemini surfactants. Many published reports have been dedicated to studying the effect of the structure of Gemini quaternary ammonium surfactants, including alkyl chain, spacer, head group, and functional group, on the interlayer environment and adsorption properties of modified Mt [25,26,27,28]. The results showed that the adsorption of phenols by modified Mt mainly occurs through hydrophobic interaction, in which the chain length and the surfactant stacking density have a synergistic effect on the adsorption capacity [29,30]. Furthermore, additional interactions derived from the functional groups of organic pollutants and Gemini surfactants, such as π–π and XH–π interactions, hydrogen bond, etc., have all been proved to have positive effects on adsorption [23,26,30].

However, up to now, several key factors affecting the application of modified Mt, such as the optimization of organic modifiers, the selection of ideal inorganic precursors, and the exploration of different adsorption mechanisms, still call for further research [30]. To the best of our knowledge, little information about the influence mechanism of the counterion of Gemini quaternary ammonium surfactants on the interlayer environment and adsorption properties of modified Mt has been published [30,31].

In this paper, Gemini quaternary ammonium surfactants with organic counterions (CH_3_CO_3_^−^ and C_6_H_5_COO^−^) and traditional halogen counterion (Br^−^) were prepared and used to modify Na–Mt. Then, the modified Mt was applied to adsorb phenol in an aqueous solution. The performance characterization of MMt–12–2–12·2CH_3_CO_3_^−^, MMt–12–2–12·2C_6_H_5_COO^−^, and MMt–12–2–12·2Br^−^ was studied in detail. The effects of some important experimental factors, such as surfactant addition, adsorbent addition, and solution pH on the adsorption performance were investigated in detail. Moreover, the adsorption kinetics and adsorption equilibrium isotherms were also discussed. More importantly, the mechanism of the influence of the counterions of the surfactants on the adsorption performance of MMT was explored.

## 2. Results and Discussion

### 2.1. Surface Activity of Gemini Quaternary Ammonium Surfactants

The surface tension values were plotted against the concentrations (*c*) of these surfactants in Figure 1. No minimum point could be found in the surface tension profiles, indicating that the purities of these synthesized Gemini surfactants were very high. Their CMC values were determined by the inflection point of the *γ*–*c* curve. The values of CMC and the surface tension at the CMC (*γ*_CMC_) were listed in Table 1. As is seen in Table 1, the CMC value of 12–2–12·2C_6_H_5_COO^−^ was the lowest among the three surfactants. The reason for this may be that the counterion of 12–2–12·2C_6_H_5_COO^−^ had a benzene ring, which increased the hydrophobic effect of the surfactant molecules in the solution, making it easier to aggregate to form micelles, resulting in a decrease in CMC.

The pC20 values of the surfactants and the standard Gibbs free energy parameters ($\Delta G_{m}^{\theta }$) of micellization are also listed in Table 1. pC20 refers to the negative logarithm of the bulk concentration of the surfactant required to reduce the surface tension of the solvent by 20 mN·m^−1^. It is an important measure of the adsorption efficiency of the surfactant. $\Delta G_{m}^{\theta }$ can be calculated from the following formula:(1)$$\Delta G_{m}^{\Theta }=RTln(\frac{CMC}{55.5})$$
where *R* represents the gas constant (8.314 J·mol^−1^·K^−1^) and *T* represents the absolute temperature (298 K).

As seen in Table 1, the order of pC20 for the three surfactants was: 12–2–12·2C_6_H_5_COO^−^ > 12–2–12·2CH_3_CO_3_**^−^** > 12–2–12·2Br^−^. For ionic surfactants, the lower the degree of hydration of the counterions, the stronger the tight binding of the hydrophilic groups, and the stronger the neutralization of the effective charges between the hydrophilic groups of the surfactant. This made the electrostatic repulsion between the surfactant ion head already adsorbed to the interface and the oncoming surfactant ion head smaller, resulting in an increase in the adsorption efficiency of the surfactant. Meanwhile, the $\Delta G_{m}^{\theta }$ values of these three surfactants were all negative, indicating that the behavior of these compounds to form micelles was spontaneous. The absolute values of $\Delta G_{m}^{\theta }$ for 12–2–12·2C_6_H_5_COO^−^, 12–2–12·2CH_3_CO_3_**^−^**, and 12–2–12·2Br^−^ decreased successively, indicating that it was increasingly unfavorable for the formation and aggregation of micelles.

### 2.2. Analysis of X-ray Diffraction

When X-ray irradiates the Mt sample, scattering occurs and can be described by the Bragg Law. According to the Bragg Law, the layer spacing of Na–Mt and MMt can be calculated. Appendix A exhibits XRD patterns of Na–Mt and MMts modified by Gemini quaternary ammonium surfactants with different counterions. As Appendix A shows, there is an obvious diffraction peak at 2θ = 6.269° in XRD curve of Na–Mt. According to the Bragg formula, the layer spacing of Na–Mt was 1.409 nm. Compared with the spectrum of Na–Mt, the diffraction peaks of MMts moved to a smaller angle, suggesting that the Gemini quaternary ammonium surfactants with different counterions had been successfully inserted into the inner space of Na–Mt, and the layer spacing of these MMts had been expanded accordingly. Meanwhile, with the increase in the amount of modifier, the layer spacing increased gradually. When the amount of modifier increased from 0.4 CEC to 2.0 CEC, the layer spacing increased significantly. When the amount of modifier increased from 2.0 CEC to 2.8 CEC, the layer spacing basically remained unchanged, or even decreased slightly. It could be inferred that when the amount of modifier was 2.0 CEC, the amount of modifier inserted between Mt layers reached saturation [32].

Figure 2A exhibited XRD patterns of Na–Mt and MMts under 2.0 CEC modification conditions. It can be seen that MMt–12–2–12·2CH_3_CO_3_^−^, MMt–12–2–12·2C_6_H_5_COO^−^, and MMt–12–2–12·2Br^−^ reached the saturation of modifier load at 2.0 CEC, while the order of the expanded layer spacing was as follows: MMt–12–2–12·2Br^−^ > MMt–12–2–12·CH_3_CO_3_^−^ > MMt–12–2–12·2C_6_H_5_COO^−^. At this point, we speculated that the counterion of the surfactant had an effect on the interlayer spacing of MMt. One of the reasons may be that the counterion of 12–2–12·2C_6_H_5_COO^−^ had a rigid-structure benzene ring, making it more difficult to enter the interlayer structure of Mt than that of 12–2–12·2Br^−^ and 12–2–12·2CH_3_CO_3_**^−^**. The other reason may be that when modified with the same concentration of the surfactant, the results from the surface tension measurements showed that 12–2–12·2C_6_H_5_COO^−^ was more likely to form micelles. At this time, there were few free surfactant monomer molecules in the solution, which hindered the ion exchange between Mt layers. In addition, as shown in Figure 2B, we speculated that the affinity of the counterions with the long aliphatic chains could also affect the interlayer spacing of MMt. Br^−^ and CH_3_CO_3_**^−^** had stronger hydrophilicity and weaker affinity with aliphatic long chains; at the same time, they failed to shield the electrostatic repulsion between the ion heads, resulting in a looser interlayer arrangement and a larger interlayer distance. On the contrary, C_6_H_5_COO^−^ had stronger hydrophobicity and stronger affinity with aliphatic long chains, which could make the arrangement of the hydrophobic long chains of surfactants tighter, resulting in stronger binding properties. This is also one of the factors that may lead to the smaller expansion of Mt interlayer spacing [33].

### 2.3. FT–IR Spectra Analysis

Figure 3 presented the FT–IR spectra of original Na–Mt, and MMt–12–2–12·2CH_3_CO_3_^−^, MMt–12–2–12·2C_6_H_5_COO^−^, and MMt–12–2–12·2Br^−^ under different CEC modification conditions. The original Na–Mt exhibited the O–H stretching vibration peak between Mt layers at 3620 cm^−1^. The peaks at 3410 cm^−1^ and 1652 cm^−1^ belonged to the stretching vibration peak and bending vibration peak of H–O–H. The bands of Na–Mt at 1095 cm^−1^ and 1017 cm^−1^ were assigned to the stretching vibration peaks of the Si–O bond, which suggested that the raw material was a silicate structure. Meanwhile, the peaks at 900~400 cm^−1^ belonged to the bending vibration peaks of Si–O and Al–O.

As shown by the b–h curves in Figure 3A, the FT–IR spectra of MMt–12–2–12·2CH_3_CO_3_^−^ under different CEC modification conditions had the characteristic absorption peaks of Na–Mt. Moreover, the infrared curves of these MMts also presented the characteristic absorption peaks of the modifiers. Among them, 3292 cm^−1^ was assigned to the stretching vibration peak of N–H bond of 12–2–12·2CH_3_CO_3_^−^, and 2922 cm^−1^ and 2853 cm^−1^ were the stretching vibration peaks of the C–H bond of –CH_3_ and –CH_2_–. Meanwhile, the peaks at 1537 cm^−1^ and 1468 cm^−1^ belonged to the bending vibration of N–H and the C–H bond of –CH_3_, respectively. The peak at 1423 cm^−1^ was the stretching vibration of carbonate in CH_3_CO_3_^−^.

The i–o and p–v curves in Figure 3B,C were the FT–IR spectra of MMt–12–2–12·2C_6_H_5_COO^−^ and MMt–12–2–12·2Br^−^, respectively. The main absorption peaks were the same as the FT–IR spectra of MMt–12–2–12·2CH_3_CO_3_^−^ and were not listed one by one. Notably, the difference was that the C=C stretching vibration peak of the benzene ring skeleton generally appeared near 1690–1500 cm^−1^, while in Figure 3B, it may coincide with the characteristic absorption peak produced by the cationic fragment of 12–2–12·2C_6_H_5_COO^−^, but it could be found that the out-of-plane bending vibration absorption peaks of C–H of the benzene ring were at 725 cm^−1^ and 695 cm^−1^. For MMt–12–2–12·2Br^−^, as shown in the p–v curves in Figure 3C, because the chemical absorption peak of the C–Br bond coincided with that of Na–Mt and the cationic fragment of 12–2–12·2Br^−^, the FT–IR spectrum only shows the characteristic absorption peaks of natural Na–Mt and the cationic fragment of 12–2–12·2Br^−^.

The above analysis presents that Gemini quaternary ammonium surfactants with different counterions have been successfully inserted into the interlayer of original Na–Mt or adsorbed on the surface of Na–Mt.

### 2.4. TG–DTG Analysis

The TG and DTG curves of original Na–Mt, and MMt–12–2–12·2CH_3_CO_3_^−^, MMt–12–2–12·2C_6_H_5_COO^−^, and MMt–12–2–12·2Br^−^ modified under 2.0 CEC are presented in Figure 4. The TG map shows that the original Na–Mt had more mass loss (13.64%) compared with MMt–12–2–12·2CH_3_CO_3_^−^ (2.32%), MMt–12–2–12·2C_6_H_5_COO^−^ (1.90%), and MMt–12–2–12·2Br^−^ (1.55%) when the temperature was lower than 200 °C. This shows that there was less free water in MMts. This is because the Gemini quaternary ammonium surfactants inserted between the Mt layers or adsorbed on the surface reduce the surface energy of Mt, and at the same time, the hydrophilic surface is converted into a hydrophobic surface. After 200 °C, the original Na–Mt had less mass loss (5.33%) compared with MMt–12–2–12·2CH_3_CO_3_^−^ (30.28%), MMt–12–2–12·2C_6_H_5_COO^−^ (28.63%), and MMt–12–2–12·2Br^−^ (30.50%). This is due to the thermal decomposition of Gemini quaternary ammonium surfactants during the heating process. It was also confirmed that the thermal stability of MMts was worse than that of the original Na–Mt [23].

The DTG curves of original Na–Mt and three kinds of MMts all had a peak at less than 200 °C, which was attributed to the detachment of physiosorbed water and free water on the surface and between layers of Mt. The original Na–Mt had a second peak at 675 °C, which was due to the dihydroxylation of –OH in the structure of Na–Mt [34,35]. These three kinds of MMts had three peaks between 200 and 600 °C, while Na–Mt did not undergo thermal decomposition in this temperature range. Therefore, the mass loss in this temperature range was caused by the thermal decomposition of the Gemini quaternary ammonium surfactants. In addition, similar to the original soil, the peak around 675 °C may be related to the dihydroxylation of –OH between Mt layers.

### 2.5. SEM Analysis

The morphologies of Na–Mt and three kinds of MMts modified under different CEC were shown in Appendix A. It was clear that the surface of the unmodified Na–Mt was flat, and the morphology was dense and stretched. From b–h, i–o, and p–v diagrams in Appendix A, we can see that after modification, the particle size and surface morphologies of MMt–12–2–12·2CH_3_CO_3_^−^, MMt–12–2–12·2C_6_H_5_COO^−^, and MMt–12–2–12·2Br^−^ samples changed to a certain extent. They presented an irregular, curly, or wrinkled surface structure in the form of loose flake aggregates, which was mainly caused by the insertion of Gemini quaternary ammonium surfactants into the silicon oxide wafer layer. Moreover, with the increase in the amount of Gemini quaternary ammonium surfactants, the surface of the three kinds of MMts was gradually warped and loose, and the fold structure was more obvious. This was due to the expansion of the layer spacing caused by the embedding of the Gemini quaternary ammonium surfactants into the interlayer of Mt wafer. With the increase in the amount of modifier, the degree of the layer spacing expansion was greater until it reached the saturation state.

### 2.6. Effect of Surfactant Addition on Phenol Adsorption by MMt

Figure 5A shows the trend of phenol adsorption by three kinds of MMts under different modifier dosages. When the amount of modifier increased from 0.4 CEC to 2.0 CEC, the adsorption capacity of MMts for phenol increased. This is because when the amount of modifier increased, the amount of the surfactant inserted into the interlayer of Mt wafer also increased, resulting in a sharp increase in the interlayer spacing of Mt, so that phenol pollutants could be adsorbed by more interlayer space. Secondly, because the surfactant was an amphiphilic molecule, the surface and interlayer of Mt modified by the surfactant were endowed with the hydrophobic groups of the surfactant, which changed the properties of Mt from hydrophilicity to hydrophobicity so as to improve the adsorption capacity of organic pollutant phenol. Therefore, when the amount of the surfactant increased, the hydrophobicity of Mt surface and interlayer also greatly increased, thus improving its adsorption capacity. When the amount of modifier increased from 2.0 CEC to 2.8 CEC, the adsorption capacity of MMts for phenol remained unchanged or even decreased. One of the reasons for this is that when the amount of surfactant increased from 2.0 CEC to 2.8 CEC, the layer spacing of MMt hardly changed, or even decreased. Another possible reason is that when the addition amount of surfactant was greater than 2.0 CEC, the additional surfactant molecules accumulated between Mt chips and occupied the active sites and interlayer space originally used for phenol adsorption, which hindered the adsorption of phenol and reduced the adsorption capacity [36,37].

At the same time, Figure 5A presents that the different counterions of the modifier had a certain impact on the adsorption of phenol by MMts: MMt–12–2–12·2Br^−^ > MMt–12–2–12·2CH_3_CO_3_^−^ > MMt–12–2–12·2C_6_H_5_COO^−^. According to the analysis results of XRD, it could be explained that the interlayer spacing of Mt was the main factor affecting the adsorption. Additionally, since C_6_H_5_COO^−^ had strong hydration, there were more water molecules around it. These water molecules could be oriented and firmly combined with ions to form a hydrophilic layer to a certain extent, thereby weakening the landing site and osmotic force of phenol molecules [38]. Therefore, the adsorption performance of MMt–12–2–12·2C_6_H_5_COO^−^ was weakened.

### 2.7. Effect of Adsorbent Addition on Phenol Adsorption by MMt

Figure 5B exhibited the adsorption effects of MMt–12–2–12·2CH_3_CO_3_^−^, MMt–12–2–12·2C_6_H_5_COO^−^, and MMt–12–2–12·2Br^−^ modified under 2.0 CEC on phenol pollutants under different dosage conditions. With the increase in the addition amount, the adsorption effect of the three kinds of MMts showed the same trend. When the amount of adsorbent was 0.04 g, the adsorption effect was best. When the amount of adsorbent increased from 0.04 g to 0.16 g, the adsorption capacity of MMts for phenol decreased gradually. This was because the more MMt was added, the number of MMt combined with phenol molecules per unit area was reduced, and the active adsorption point of MMt was not saturated, which reduced the adsorption efficiency of MMt. Secondly, under the current concentration and volume of the phenol solution, with the increase in the amount of MMt and the increase in the ratio of solid to liquid, the viscosity of the system increased and the dispersion of MMt became worse, so that MMt could not fully contact with phenol molecules. As a result, a part of MMt could not display full performance to its own adsorption [39].

### 2.8. Effect of pH on Phenol Adsorption by MMt

It is well known that solution pH has an impact on the surface charge of the adsorbent and ionization degree of adsorbate. Figure 5C shows the adsorption effects of MMt–12–2–12·2CH_3_CO_3_^−^, MMt–12–2–12·2C_6_H_5_COO^−^, and MMt–12–2–12·2Br^−^ modified under 2.0 CEC on phenol pollutants under different pH conditions. As shown in Figure 5C, when the pH of the system decreased from 8 to 2, the adsorption capacity of three kinds of MMts decreased, while as the pH value increased from 8 to 10, their adsorption capacities exhibited an increasing trend. It is well known that the existing form of phenols at different pH and the surface charge of MMt account for this phenomenon. The pKa of phenol is 9.96. When the pH of the system was 2~9, the phenol presented in a molecular form in the solution, and phenol was adsorbed to the surface of Mt through physical adsorption. When the pH of the system increased to 10, the phenol in the solution dissociated into phenol anion. Meanwhile, the phenol anion combined with the residual positive charge on the surface of Mt and the N-positive ion in the surfactant between Mt layers, which greatly increased the adsorption efficiency of MMt [3,5]. Therefore, the optimum adsorption effect could be obtained when the pH was 10, and the adsorption capacities of MMt–12–2–12·2CH_3_CO_3_^−^, MMt–12–2–12·2C_6_H_5_COO^−^, and MMt–12–2–12·2Br^−^ could reach 100.834 mg/g, 99.985 mg/g, and 115.110 mg/g, respectively. Compared with the adsorption capacities of the original Na–Mt, which was only 66.120 mg/g, the adsorption capacities of all three kinds of MMts improved greatly.

### 2.9. Adsorption Kinetics

As presented in Figure 6, at 25 °C, 35 °C, and 45 °C, the adsorption amount of phenol by three kinds of MMts increased with time and reached equilibrium in approximately 120 min. Compared with MMt–12–2–12·2Br^−^ and MMt–12–2–12·2CH_3_CO_3_^−^, the time for MMt–12–2–12·2C_6_H_5_COO^−^ to reach adsorption equilibrium was slightly longer. The adsorption rate increased rapidly in first 30 min and then reached equilibrium gradually because there were abundant adsorption surface sites at the beginning, and less and less vacant sites were available for adsorption with increasing the contact time. In addition, the remaining vacant sites were hard occupy due to the increasing electrostatic repulsion between the phenol molecule adsorbed on the surface of MMt and that in the solution. It can also be seen in Figure 6 that the adsorption effect of the three kinds of MMts decreased slightly with the reaction temperature increasing from 25 °C to 45 °C. This is because the adsorption of phenol by MMT was an exothermic process. In order to ensure adsorption desorption equilibrium and the best adsorption effect, the reaction time increased to 6 h and the reaction temperature was set at 25 °C.

Subsequently, the experiment data of adsorption kinetics at 25 °C were fitted by the pseudo-first-order model (Equation (2)), pseudo-second-order model (Equation (3)), and intra-particle diffusion model (Equation (4)):(2)$$ln\left(q_{e}-q_{t}\right)=lnq_{e}-k_{1}t$$
(3)$$\frac{t}{q_{t}}=\frac{1}{k_{2}q_{e}^{2}}+\frac{t}{q_{e}}$$
(4)$$\;q_{t}=k_{p}t^{1/2}+C$$
where *q_e_* (mg/g) is the equilibrium adsorption capacity of MMt, *q_t_* (mg/g) stands for the amount of phenol adsorbed by MMt at time *t* (min), and *k*_1_ (g/(mg·min)^−1^) and *k*_2_ (g/(mg·min)^−1^) represent the rate constants of two kinetic models. *k_p_* (mg/g·min^1/2^) is the intra-particle diffusion model constant, and *C* represents the intra-particle diffusion rate constant.

The adsorption performance of MMt–12–2–12·2CH_3_CO_3_^−^, MMt–12–2–12·2C_6_H_5_COO^−^, and MMt–12–2–12·2Br^−^ modified under 2.0 CEC were analyzed by three kinetic models, and the kinetic fit curves are shown in Figure 7 and Figure 8. The relevant kinetic parameters are presented in Table 2. As given in Table 2, the *R*^2^ values of the pseudo-second-order model were higher than those of the pseudo-first-order model, and the values of *q_e cal_*, obtained from the pseudo-second-order model, were closer to *q_e exp_*, which demonstrated that the adsorption kinetic data of MMt–12–2–12·2CH_3_CO_3_^−^, MMt–12–2–12·2C_6_H_5_COO^−^, and MMt–12–2–12·2Br^−^ were more fitted to the pseudo-second-order model [40].

According to the three linear regions fitted by the intra particle diffusion model (Figure 8), the adsorption process of phenol on MMt could be divided into three stages. The first stage was the outer surface adsorption stage; that is, the diffusion of phenol molecules in the system to the outer surface of MMt, which was a transient adsorption process. The fitting line in the first stage was the steepest and the slope was the highest, indicating that the adsorption rate in this stage was fast. The second stage was the stepwise adsorption stage, and the diffusion rate of the phenol molecule itself dominated the whole adsorption process. The third stage was the equilibrium adsorption stage. Compared with the fitting curve in the first two stages, the steepness of the fitting straight line in the third stage was greatly reduced because the diffusion process in the particles was gradually slowed down due to the decrease in adsorbate concentration and the active adsorption point in the process.

According to the relevant dynamic parameters listed in Table 3, the fitting of the three stages of the intra particle diffusion models of MMt–12–2–12·2CH_3_CO_3_^−^, MMt–12–2–12·2C_6_H_5_COO^−^, and MMt–12–2–12·2Br^−^ were linear lines, but the line segments of all stages did not pass through the origin (*C* ≠ 0), which proved that the intra-particle diffusion model was not the only mechanism controlling the adsorption process, and other adsorption mechanisms were also involved, for example, external liquid film diffusion, surface diffusion, and distribution. In addition, the *C* value of the second stage of the MMt–12–2–12·2Br^−^ intra-particle diffusion model was the highest, indicating that the MMt–12–2–12·2Br^−^ boundary layer had the largest thickness and the phenol molecules were easy to diffuse inside the particles, which may also be one of the reasons for the maximum adsorption capacity of MMt–12–2–12·2Br^−^. The correlation coefficient *R*^2^ of the second linear region of the three MMts was higher than 0.93, which presented that the intra-particle diffusion process played an important role in the adsorption process [41,42].

### 2.10. Adsorption Equilibrium Isotherms

In this study, Langmuir and Freundlich models were chosen to describe the adsorption mechanism:(5)$$lnq_{e}=lnk_{F}+\frac{lnC_{e}}{n}$$
(6)$$\frac{C_{e}}{q_{e}}=\frac{1}{q_{max}K_{L}}+\frac{C_{e}}{q_{max}}$$
where *q_max_* (mg/g) is the maximum adsorption capacity, *q_e_* (mg/g) and *C_e_* (mg/L) have the same definitions as above, *K_L_* (L/mg) is the adsorption isotherm constant of Langmuir, and *K_F_* and *n* are the Freundlich constants.

The adsorption isotherm fitting plots of MMt–12–2–12·2CH_3_CO_3_^−^, MMt–12–2 12· 2C_6_H_5_COO^−^, and MMt–12–2–12·2Br^−^ are exhibited in Figure 9. The corresponding parameters’ estimated values are exhibited in Table 4. It is clear that the adsorption of the three kinds of MMts were all better fitted with the Freundlich rather than the Langmuir adsorption isotherm model. For MMt–12–2–12·2CH_3_CO_3_^−^, MMt–12–2–12·2C_6_H_5_COO^−^, and MMt–12–2–12·2Br^−^, the linear correlation coefficients R^2^ of the Freundlich model were 0.996, 0.997, and 0.969, respectively, which indicates that phenol was adsorbed in different MMt in a multilayer arrangement.

As given in Table 4, the exponent n of MMt–12–2–12·2C_6_H_5_COO^−^ was slightly less than 1, while the n values of MMt–12–2–12·2CH_3_CO_3_^−^ and MMt–12–2–12·2Br^−^ were greater than 1, indicating that these adsorption processes were spontaneous reactions and physical processes, and the adsorption process of MMt–12–2–12·2CH_3_CO_3_^−^ and MMt–12–2–12·2Br^−^ occurred more strongly than that of MMt–12–2–12·2C_6_H_5_COO^−^ [43].

### 2.11. Thermodynamic Parameters

Gibbs free energy change (Δ*G*°), enthalpy change (Δ*H*°), and entropy change (Δ*S*°), which were calculated by Equations (7) and (8) and presented in Table 5, could provide in-depth information on the internal energy changes during the adsorption process. In Equations (7) and (8), R is the universal gas constant, *T* is the absolute temperature, and *K*(= *q_e_/C_e_*) represents the distribution coefficient. The values of Δ*H*° and Δ*S*° can be determined by the intercept and slope of the linear plot of lnK versus 1/*T*.
(7)lnK=ΔS°/R−ΔH°/RT
(8)ΔG°=ΔH°−TΔS°

The negative values of Δ*G*° at 298, 308, and 318 K, Table 5, indicate the feasibility of phenol adsorption onto three kinds of MMTs and the spontaneous nature of the adsorption process. Furthermore, in theory, the physical sorption process was valid when the Δ*G*° values were between −20 and 0 KJ·mol^−1^, and the chemical sorption process was generally valid for Δ*G*° between −400 and −80 KJ·mol^−1^ [44]. It can be seen that the Δ*G*° values were closer to the realm of physisorption, indicating that the phenol adsorption onto these MMTs was mainly physical adsorption. In addition, for these three MMTs, the absolute values of Δ*G*° increased with the temperature decreasing, resulting in a higher adsorption capacity. Additionally, the negative value of Δ*H*° confirmed that the adsorption process was exothermic, following the phenol equilibrium sorption capacity decreasing with increasing solution temperature. The negative value of Δ*S*° indicated a decrease in the randomness at the solid/liquid interface during the adsorption process [45].

## 3. Materials and Methods

### 3.1. Materials

The original Na–Mt was provided by Beijing Innochem Technology Co., Ltd. Phenol (≥99%) was purchased from Shanghai Aladdin Biochemical Technology Co., Ltd. The high-purity water (ρ = 18.25 MΩ·cm) with a surface tension of 72 mN·m^−1^ was supplied by an ultrapure laboratory water purification system. Three kinds of Gemini quaternary ammonium surfactants (12–2–12·2CH_3_CO_3_^−^, 12–2–12·2C_6_H_5_COO^−^, 12–2–12·2Br^−^), the structure of which is shown in Figure 10, were prepared according to our previous paper and some other studies [1,46,47].

### 3.2. Surface Tension Measurements of Gemini Quaternary Ammonium Surfactants

At 25 °C, the surface tension of the surfactant aqueous solutions within a concentration range of 10^−6^~10^−3^ mol·L^−1^ was measured using the Wilhemy plate method on a Dataphysics tensiometer DCAT11. All solutions were prepared with the high-purity water. Each sample was measured three times. Surface tension was an average of triplicate measurements, and the standard deviation was less than 0.01.

### 3.3. Preparation of MMt

The CEC of the original Na–Mt sample determined by the ethanol–ammonium chloride method was 70 mmol/100 g. The samples of MMt were prepared by the following steps. The raw material of Na–Mt (2 g) was placed into 100 mL ultrapure water with constant stirring. Then, we added the corresponding mass of modifier calculated according to Equation (9) into another 100 mL deionized water. According to the CEC of Na–Mt, modifier solutions of 0.4, 0.8, 1.2, 1.6, 2.0, 2.4, and 2.8 CEC were prepared, and then were added to the Na–Mt suspension. After all were transferred, the products were sealed and oscillated for 24 h. After the reaction, the solution was filtered and eluted with a large amount of ultrapure water. When the surface tension of the washed solution was consistent with that of ultrapure water, the obtained solid samples were dried at 100 °C for 24 h. After full drying, MMts modified with 12–2–12·2CH_3_CO_3_^−^, 12–2–12·2C_6_H_5_COO^−^, and 12–2–12·2Br^−^ were ground into powder by mortar and denoted as MMt–12–2–12·2CH_3_CO_3_^−^, MMt–12–2–12·2C_6_H_5_COO^−^, and MMt–12–2–12·2Br^−^, respectively.

The amounts of modifier can be evaluated by the following equation:(9)$$m_{s}=n\times CEC\times M\times m_{0}$$
where $m_{s}$(g) is the amount of modifier added, n represents the multiple of CEC, CEC (mmol/L) stands for the cation exchange capacity of Na–Mt, M is the relative molecular weight of the modifier, and $m_{0}$(g) is the mass of the Na–Mt in the Na–Mt suspension prepared in the previous step.

### 3.4. Characterization of MMt

XRD patterns of native Na–Mt and MMts were recorded by X-ray diffractometer (D8 Advance, Bruker) with Cu Kα target radiation (λ = 0.15406 nm, Tube voltage = 40 KV, Tube current = 40 mA, 2θ = 2°–10°, Scanning speed = 1°/min). The divergent slit, anti-divergent slit, and receiving slit were set to 1 mm, 1 mm, and 0.1 mm, respectively. FT–IR spectra of the samples were analyzed by FT–IR spectrometer (Nicolet ISLO, Thermo Fisher), scanning from 4000 cm^−1^ to 400 cm^−1^. The thermogravimetric (TG)/derivative thermogravimetric (DTG) analysis of native Na–Mt and MMt were gained by simultaneous thermal analyzer (METTLER TGA/DSC3+) in a high-purity nitrogen flow (99.99%) at a flow rate of 40 mL/min, and the sample was heated from 40 °C to 900 °C at a heating rate of 20 K/min. The morphology of Na–Mt and MMts were observed with SEM (Carl Zeiss Sigma300, Carl Zeiss) at the voltage of 5.00 kV and the magnification of 2.50 KX.

### 3.5. Adsorption Experiments

The adsorption capacities of three kinds of MMts for phenol were carried out by a series of adsorption experiments. The influence of varying factors including surfactant addition, adsorbent addition, and solution pH on the uptake of phenol pollutant by three kinds of MMts were investigated in detail. All experiments were conducted in triplicate.

To determine the concentration of the surfactant, 0.04 g MMt modified with different doses of surfactants was added into 100 mL of phenol solution with an initial concentration of 100 mg/L. The dispersions were sealed and oscillated for 6 h at 25 °C, and then were filtered through a 0.45 μm membrane filter. After that, the concentrations of the residual phenol were determined using the UV–vis near-infrared spectrometer (UV–3600, SHIMADZU, Japan) with the analytical wavelength of 270 nm (pH = 2~9) and 286 nm (pH = 10). The adsorption amounts of phenol on adsorbents can be evaluated by the following equation:(10)$$\;q_{e}=\frac{\left(C_{0}-C_{e}\right)*V}{m}$$
where *q_e_* (mg/g) is the adsorption capacity of MMt for phenol, *C*_o_ (mg/L) represents the initial concentration of phenol, *C_e_* (mg/L) is the equilibrium concentration of phenols, *v* (L) stands for the volume of phenol solution, and *m*(g) is the mass of MMt.

To determine the adsorbent addition, 0.04 g, 0.06 g, 0.08 g, 0.10 g, 0.12 g, 0.14 g, and 0.16 g MMt was added into phenol solution. To determine the pH value of the solution, the pH was adjusted to 2~10 by 0.1 mol/L HCl solution and 0.1 mol/L NaOH solution. For all the experiments related to the effect of adsorbent addition and pH, the initial concentration and amount of phenol were 100 mg/L and 100 mL, the dispersions were sealed and oscillated for 6 h at 25 °C.

To study adsorption kinetics, 0.04 g MMt modified at 2.0 CEC was added into 100 mL of phenol solution with an initial concentration of 100 mg/L and the dispersions were oscillated for 7 h at 25 °C, 35 °C, and 45 °C. At different intervals, an aliquot of the reaction solution was quickly sampled and detected.

Adsorption isotherm was obtained using batch experiments; 0.04 g MMt modified at 2.0 CEC was added into 100 mL of phenol solution with different initial concentrations of 20, 50, 80, 100, and 150 mg/L. The dispersions were sealed and oscillated for 6 h at 25 °C.

## 4. Conclusions

In this study, phenol was removed from water using MMt modified by Gemini quaternary ammonium surfactants containing different counterions. It is worth noting that the counterions affected the properties of the surfactants, which, in turn, affected the interlayer spacing and adsorption properties of MMt. All three kinds of MMts reached the optimum adsorption capacity under the conditions of the saturated intercalation concentration 2.0 CEC of the original Na–Mt, 0.04 g of adsorbent, and a pH = 10, but there were differences in the adsorption capacity among different MMt, which was MMt–12–2–12·2Br^−^ > MMt–12–2–12·2CH_3_CO_3_^−^ > MMt–12–2–12·2C_6_H_5_COO^−^. We speculated that the rigid structure of C_6_H_5_COO^−^ made it more difficult for 12–2–12·2C_6_H_5_COO^−^ to enter the interlayer structure of Mt due to the poor hydrophilicity of C_6_H_5_COO^−^. 12–2–12·2C_6_H_5_COO^−^ was more likely to form micelles and the ion exchange between Mt layers was hindered, thus causing a smaller expansion of Mt interlayer spacing. In addition, C_6_H_5_COO^−^ had strong hydration, which weakened the landing site and osmotic force of phenol molecules, resulting in a poor adsorption effect. On the contrary, due to the strong hydrophilicity of Br^−^ and CH_3_CO_3_^−^, there were more surfactant monomer molecules in the aqueous solution, which promoted the ion exchange. Meanwhile, the affinity between the long aliphatic chain and the counterions (Br^−^ and CH_3_CO_3_^−^) existing in the Mt layer was weak and failed to shield the electrostatic repulsion between the ion heads, resulting in a looser arrangement between the layers and a larger interlayer spacing. Moreover, the adsorption kinetics of all adsorption process was in good agreement with the pseudo-second-order kinetics model, and the adsorption isotherms were better modeled by the Freundlich isotherm. The thermodynamic parameters indicated that the adsorption of phenol was a physical, spontaneous, and exothermic process.

Compared with other modifiers, 12–2–12·2CH_3_CO_3_^−^ was synthesized using a green chemical material–dimethyl carbonate. Although the adsorption capacity of MMt–12–2–12·2CH_3_CO_3_^−^ was slightly smaller than that of MMt–12–2–12·2Br^−^, it can effectively avoid the secondary pollution of the environment in practical adsorption applications. Therefore, we conclude that MMt–12–2–12·2CH_3_CO_3_^−^ can serve as a green and potential adsorbent with high efficiency in the treatment of contaminants in wastewater.

## Figures and Tables

**Figure 1 molecules-28-02021-f001:**
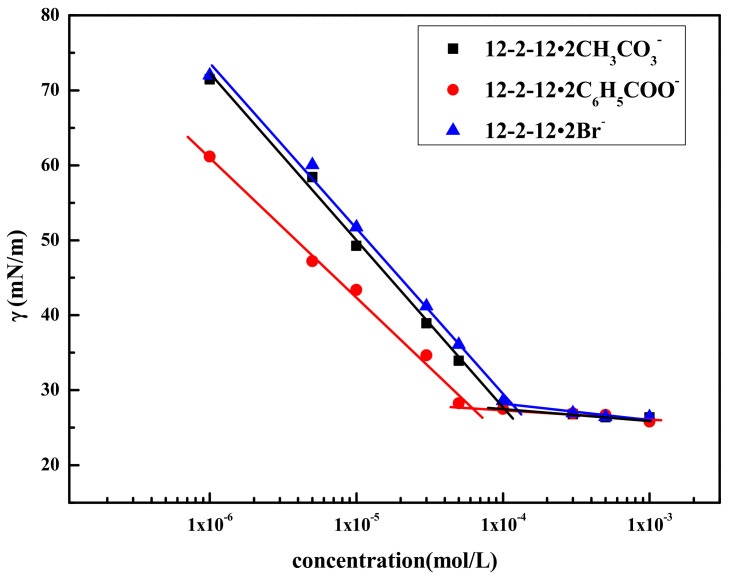
Plots of *γ* against the concentration of 12–2–12·2CH_3_CO_3_^−^, 12–2–12·2C_6_H_5_COO^−^, and 12–2–12·2Br^−^ at 25 °C.

**Figure 2 molecules-28-02021-f002:**
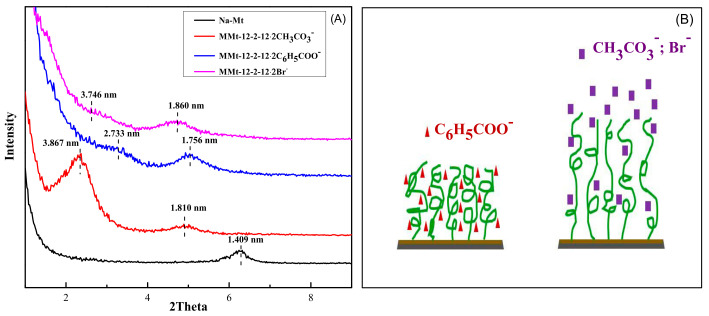
(**A**): XRD patterns of Na–Mt, and MMt–12–2–12·2CH_3_CO_3_^−^, MMt–12–2–12· 2C_6_H_5_COO^−^, and MMt–12–2–12·2Br^−^ under 2.0 CEC modification conditions; (**B**): Schematic diagram of interaction between counterions and the long aliphatic chains of the surfactants in Mt layers.

**Figure 3 molecules-28-02021-f003:**
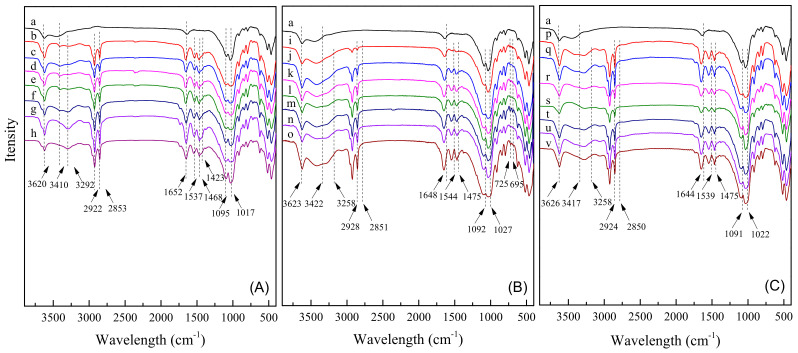
FT–IR patterns of Na–Mt (a), and MMt–12–2–12·2CH_3_CO_3_^−^ (**A**), MMt–12–2–12·2C_6_H_5_COO^−^ (**B**), and MMt–12–2–12·2Br^−^ (**C**) under different CEC modification conditions [0.4 CEC (b, i, p), 0.8 CEC (c, j, q),1.2 CEC (d, k, r),1.6 CEC (e, l, s), 2.0 CEC (f, m, t), 2.4 CEC (g, n, u), 2.8 CEC (h, o, v)].

**Figure 4 molecules-28-02021-f004:**
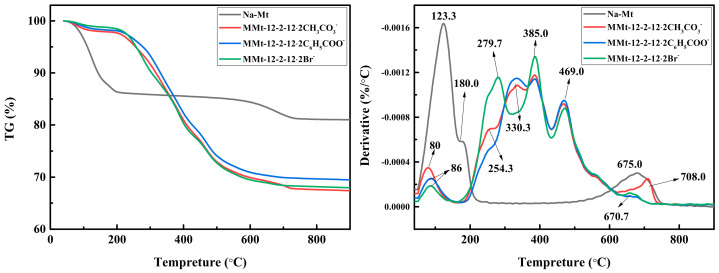
TG and DTG curves of Na–Mt and MMts modified under 2.0 CEC.

**Figure 5 molecules-28-02021-f005:**
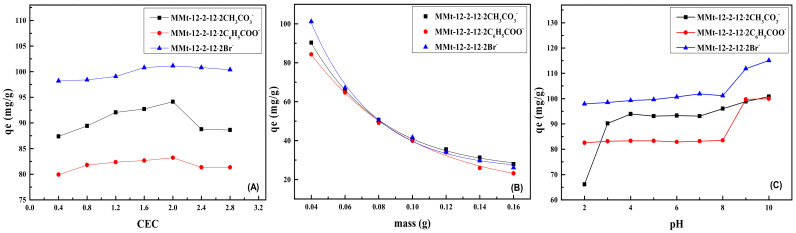
Effect of modifier dosage (**A**), adsorbent addition (**B**), and pH (**C**) on phenol adsorption by MMts.

**Figure 6 molecules-28-02021-f006:**
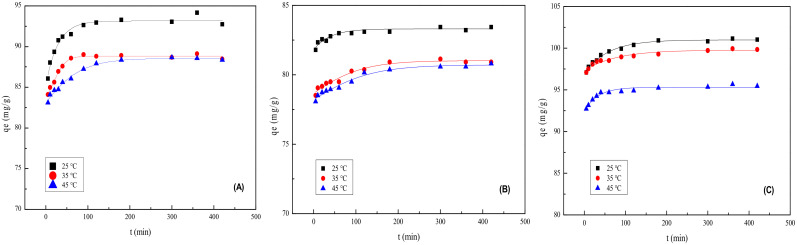
Effect of reaction time and reaction temperature on phenol adsorption by (**A**): MMt–12–2–12·2CH_3_CO_3_^−^, (**B**): MMt–12–2–12·2C_6_H_5_COO^−^, and (**C**): MMt–12–2–12·2Br^−^.

**Figure 7 molecules-28-02021-f007:**
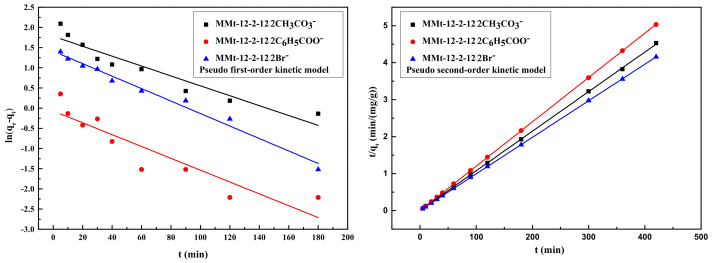
Pseudo-first-order kinetic model and pseudo-second-order kinetic model of phenol adsorption on three kinds of MMTs modified under 2.0 CEC at 25 °C.

**Figure 8 molecules-28-02021-f008:**
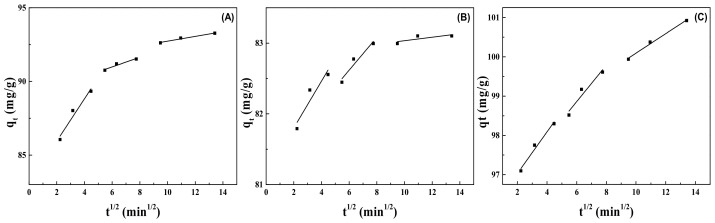
Intra-particle diffusion model of phenol adsorption on MMt modified under 2.0 CEC [MMt–12–2–12·2CH_3_CO_3_^−^ (**A**), MMt–12–2–12·2C_6_H_5_COO^−^ (**B**), MMt–12–2–12·2Br^−^ (**C**)] at 25 °C.

**Figure 9 molecules-28-02021-f009:**
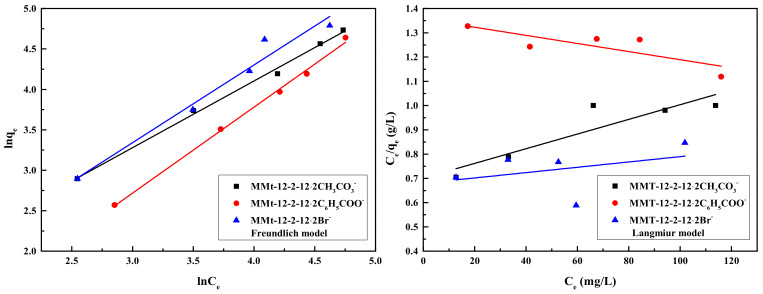
Freundlich model and Langmuir model for adsorption of phenol pollutants on MMt modified under 2.0 CEC at 25 °C.

**Figure 10 molecules-28-02021-f010:**
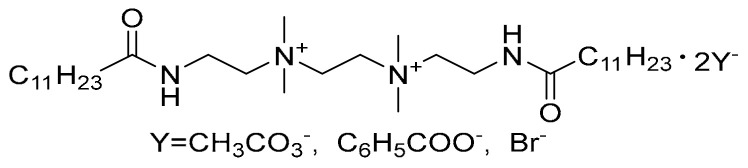
The structure of Gemini quaternary ammonium surfactants.

**Table 1 molecules-28-02021-t001:** Surface property parameters of Gemini quaternary ammonium surfactants at 25 °C.

Compound	CMC	*γ_CMC_*	pC20	$\Delta G_{m}^{\theta }$
	(mol·L^−1^)	(mN·m^−1^)	(mol·L^−1^)	(KJ·mol^−1^)
12–2–12·2CH_3_CO_3^−^_	1.04 × 10^−4^	27.65	3.983	−32.69
12–2–12·2C_6_H_5_COO^−^	6.20 × 10^−5^	27.52	4.208	−38.49
12–2–12·2Br^−^	1.19 × 10^−4^	27.93	3.924	−38.49

**Table 2 molecules-28-02021-t002:** Pseudo-first-order and pseudo-second-order kinetic model parameters of phenol adsorption by MMt modified under 2.0 CEC.

MMt	*q_e exp_* (mg/g)	Pseudo-First-Order Model			Pseudo-Second-Order Model		
		*k* _1_	*q_e cal_*	*R* ^2^	*k* _2_	*q_e cal_*	*R* ^2^
MMt–12–2–12·2CH_3_CO_3_^−^	94.140	0.01226	5.923	0.90843	0.01438	93.458	0.99992
MMt–12–2–12·2C_6_H_5_COO^−^	83.212	0.01467	0.933	0.84495	0.04016	83.403	1
MMt–12–2–12·2Br^−^	101.139	0.01546	4.102	0.98400	0.01417	101.215	0.99999

**Table 3 molecules-28-02021-t003:** Intra-particle diffusion model parameters of phenol pollutants adsorbed by MMt modified under 2.0 CEC.

MMt	Intra-Particle Diffusion Model		
	*k_p_*	*C*	*R* ^2^
MMt–12–2–12·2CH_3_CO_3_^−^	0.32625	89.02926	0.94919
MMt–12–2–12·2C_6_H_5_COO^−^	0.23190	81.22691	0.93390
MMt–12–2–12·2Br^−^	0.46000	96.07646	0.93390

**Table 4 molecules-28-02021-t004:** Adsorption isotherm model parameters of phenol pollutants adsorbed by MMt modified under 2.0 CEC.

MMt	Freundlich Adsorption Isotherm			Langmuir Adsorption Isotherm		
	*k_F_*	*N*	*R* ^2^	*k_L_*	*q_max_*	*R* ^2^
MMt–12–2–12·2CH_3_CO_3_^−^	2.236	1.212	0.996	0.00431	331.126	0.829
MMt–12–2–12·2C_6_H_5_COO^−^	0.623	0.940	0.997	0.00123	598.802	0.670
MMt–12–2–12·2Br^−^	1.581	1.040	0.969	0.00162	909.091	0.143

**Table 5 molecules-28-02021-t005:** Thermodynamic parameters of the phenol adsorption onto MMT.

MMT	*C*_0_ Mg·L^−1^	Δ*H*° kJ·mol^−1^	Δ*S*° J·mol^−1^·K^−1^	Δ*G*° (KJ·mol^−1^)			*R* ^2^
				298K	308K	318K	
MMt–12–2–12·2CH_3_CO_3_^−^	100	−3.380	−8.122	−0.959	−0.878	−0.797	0.847
MMt–12–2–12·2C_6_H_5_COO	100	−1.967	−4.819	−0.531	−0.483	−0.435	0.904
MMt–12–2–12·2Br^−^	100	−3.736	−8.105	−1.321	−1.240	−1.159	0.934

## Data Availability

Not applicable.

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
