# Peer review of "Organo-Montmorillonite Modified by Gemini Quaternary Ammonium Surfactants with Different Counterions for Adsorption toward Phenol"

_molecules, 2023, doi:10.3390/molecules28052021_

Round 1

Reviewer 1 Report

Manuscript entitled “Organo-montmorillonite modified by Gemini quaternary ammonium surfactants with different counterions for adsorption toward phenol” can be accepted for publication in the Molecules Journal.

The novelty and practical applicability of this study are clearly highlighted in the manuscript.

Here is a list of my specific comments:

1-      What does the abbreviation “CEC” mean in the  Preparation of MMt section?

2-      To better understand the effect of pH conditions on the adsorption process, the authors are invited to identify the PZC of MMt-12-2-12·2CH3CO3-, MMt-12-2-12·2C6H5COO- and MMt-12-2-12·2Br- (section “Effect of pH on phenol adsorption by MMt”)

3-      Please read the references

Journal of Separation and Purification Technology 251(2021)117335.

Journal of Molecular Liquids, 335(2021)116560

Reviewer 2 Report

The topic is interesting in the context of environmental protection and a lot of work has been already done with layered minerals. The authors presented a research report, although the problem of further utilization of the adsorbent after the purification process is interesting. I hope that there will also be an article that will explain these issues, compared to activated carbons, for example.
The adsorption testing methodology is not exhaustively described in the "Materials and methods" section. In particular, the method of examining the adsorption kinetics and adsorption isotherms is not described. Adsorption specialists probably don't have a problem with this - we all know how it's done - but for related professionals, it may be less understandable. I would suggest adding a research procedure for adsorption kinetics and adsorption isotherms (especially the waiting time for establishing the adsorption equilibrium is important).

Suggested detailed changes or comments to the text in respective lines:
20 - 21 Authors should decide to use the 'counterion' or 'counter ion' now that it is written as one or two words. In the following text it is one-word.
73 - 75 What the authors meant? That sentence should be rebuilt.
112 and previous - what means slow addition? The liquid was dosed by a peristaltic pump or the volume was divided into 2, 5, or 10 portions and added consecutively. It is interesting if stirring velocity influences the final product. It would be better to describe more precisely that different concentrations of surfactant were added to successive batches of the suspension. Otherwise, it is left to the reader's discretion.
124  probably it should be: '...the mass of the Na-Mt in suspension...'. Do I understand well?
128 Misspelled name, "Bruker" is correct.
144 Proposed change: "...was added into several 100 mL.."
158 Proposed change: "...was the quantity of MMt."
307 Figures S2-S4 cited in SEM analysis chapter are not included in the file received for review. So, their discussion is not illustrated.

The pdf file with highlights is attached.

Reviewer 3 Report

The present study was focused on the phenol adsorption onto modified-Montmorillonite with GEMINI Quaternary 2 Ammonium Surfactants. In spite of a few novel and original findings, it cannot be accepted for the publication unless it will be revised. See the following comments.

1.     The mechanistic consideration of phenol adsorption in the presence of modifiers seems remarkably poor. We can understand the effect of modifiers onto the adsorption behaver, but nobody see how the adsorption occur. What is the key step?

2.     What is the role of counterions? Acidity? Structure? Or?

3.     The authors should make the behavior of phenol much clearer. Maybe, phenol is not fully dissociated but is in equilibrium in the solution phase. Phenol may be dissociated on the surface of modified-Montmorillonite as a result of adsorption.

Round 2

Reviewer 3 Report

The revised version is not always satisfactory, but the author has responded politely to my comments. Along with their correction, I am sure, it will be accpeted for the publication.